# Ultrasound Guided Surgery as a Refinement Tool in Oncology Research

**DOI:** 10.3390/ani12233445

**Published:** 2022-12-06

**Authors:** Juan Antonio Camara Serrano

**Affiliations:** Preclinical Therapeutics Core, University of California San Francisco, San Francisco, CA 94158, USA; juanantonio.camaraserrano@ucsf.edu; Tel.: +1-628-629-3555

**Keywords:** preclinical models, oncology, ultrasounds injections, refinement

## Abstract

**Simple Summary:**

This review describes the potential use of ultrasound as a refinement tool in research. After a general state-of-the-art discussion, the most frequent organs used as a host for orthotopic models in oncology research are listed, including the thyroid gland, heart, liver, spleen, kidney, pancreas, uterus, and testicles. In each organ and after a short ultrasonography description, the practical protocol for the ultrasound-guided injection is described as well as the main risk of the procedure and technical limitations. The main objective of this work is to help users with the use of ultrasound-guided injection. For this purpose, the descriptions of the protocols are mainly practical with tips which are frequent mistakes carried out during the injections.

**Abstract:**

Refinement is one of the ethical pillars of the use of animals in research. Ultrasonography is currently used in human medicine as a surgical tool for guided biopsies and this idea can be applied to preclinical research thanks to the development of specific instruments. This will eliminate the necessity of a surgical opening for implanting cells in specific organs or taking samples from tissues. The approach for the injection will depend on the target but most of the case is going to be lateral, with the probe in a ventral position and the needle going into from the lateral. This is the situation for the thyroid gland, heart, liver, spleen, kidney, pancreas, uterus, and testicles. Other approaches, such as the dorsal, can be used in the spleen or kidney. The maximum injected volume will depend on the size of the structure. For biopsies, the technical protocol is similar to the injection knowing that in big organs such as the liver, spleen, or kidney we can take several samples moving slightly the needle inside the structure. In all cases, animals must be anesthetized and minimum pain management is required after the intervention.

## 1. Introduction

The 3Rs principle was enunciated in the early years of the 60s of the past century by Russell and Burch, two English biologists, in their book “The principle of Humane Experimental Technique”. This publication was an outbreak in the manipulation of lab animals, that until this work, were basically considered a research tool such as an Eppendorf tube or a pipette [1]. In their work, Russell and Burch defined the ethical principles that must rule over any experimental work which involves animals, especially the ones with a more developed neurological system, such as mammals or birds. In recent years, these ethical principles cover other animals such as octopuses, which have been raised as conscious and sensible to pain subjects, at a higher level that could be assumed due to their phylogenetic stratum [2].

The concepts included in the 3Rs principle are replacement, reduction, and refinement. The first one refers to the ethical command to use non-in vivo methodologies for research when accessible, for example, using in vitro studies or in silico experiments. The second concept, reduction, commits to using fewer animals as possible. The total number of animals included in an experiment must be justified and statistical analysis must be run to calculate the minimum sample size for every experimental group [3]. The third concept, refinement, stands for the development of less invasive techniques and the reduction in the damage done to the animals during the research work.

Looking at the current scenario, the reduction principle has been propelled in the last years with big data tools and artificial intelligence that make it possible to obtain trustable conclusions from experiments with only a single patient or specimen [4]. In the same way, replacement is taking advantage of the astonishing development of computer science and nowadays we can design algorithms for simulating viral replications, drug effects in cells, or cancer cell mutations [5]. Perhaps, refinement has been the less developed concept since the beginning of the 3Rs principle. We still need to manipulate the animals and in most cases create some sort of disconformity to them. Even so, the minimum required welfare levels are much higher than in the past decades, taking advantage, for example, of the advances in analgesia and anesthesia or the new surgical techniques, less invasive and more refined [6,7].

Oncology research is one of the most important in medicine, due to the global economic impact of these pathologies, as well as the emotional charge that “the big C” generates in the patient and the families. In oncology research, animal models are a basic part of the process, not only for drug discovery but also because we do not still have full knowledge of the process that drives a cell to become a cancerous entity. In both basic research and drug discovery, animal models are a pillar in the projects. In some of these works, the animals get implanted with a sample of human tumor tissue, or tumoral cells, and a surgical procedure is required for this implantation. Frequently, these implantations are performed in the subcutaneous tissue, but in other cases, the implantation is performed in different locations, and we talk about orthotopic models, where the tumor cells are implanted in the physiological host organ [7,8]. A clarifying example can be the pancreatic implantation of a human pancreatic cancer sample, or the intravascular injection of blood cancer cells. These orthotopic models are, in theory, more similar to the clinical scenario than the subcutaneous models because the host tissue is similar to the original one [7].

The use of imaging techniques for preclinical research dated from several decades ago, when lab animals were scanned in clinical devices for obtaining information about the internal structures of the animals, due to the absence of specific technologies designed for preclinical research. However, in recent years, we witnessed the arising of different technology for lab animals, including specific instruments such as microPET (positron emission tomography), microCT (computed tomography), MRI (magnetic resonance imaging), optical imaging, or ultrasonography. All these technologies play a significant role in the application of the 3Rs principle in cancer models. The possibility to look into the animals avoiding their sacrifice represents a dual effect on animal welfare: on one side the necessity to euthanize the animal is eliminated (refinement) and on the other side we can rescan the same animal at different time points during the experiment (reduction). Furthermore, repeating the imaging on the same animal allows to use them as their own control, which increases the statistical significance of the experiments. Resuming, the imaging technologies development had a significant impact on animal welfare in preclinical research [7].

Specifically talking about ultrasonography, new equipment with higher ultrasound frequencies which increase the spatial and temporal resolution make it possible to obtain images that will be not accessible from a clinical device designed for human patients and sizes. Ultrasonography has a significant advantage which is real-time imaging. What we see on the monitor is happening inside the patient. This opens a door to the use of this technology as a surgical tool for accessing internal structures in the animal. We can make injections or take biopsies avoiding conventional surgical procedures, which will reduce the requirements of post-surgical care, due to the minimum damage created and the low pain generated to the patient/animal. Ultrasonography-guided surgery has been performed for years in human medicine and pets and now, with specific equipment for lab animals, it is time for including it as a refinement tool in animal models and preclinical research.

## 2. State of the Art

When talking about preclinical imaging, it is necessary to make a difference between animal models. In big animals, above the size of a rabbit, clinical imaging devices have been used for research for several decades, due to the similar size of the animals to the human patients and the acceptable quality of the images. This did not happen in lab animals. The maximum spatial resolution of the clinical equipment was not enough to get acceptable images from an animal that could weigh, in the worst case, less than 20 grams. During the last years, a significant improvement in imaging technologies has made an impact on animal research. Specific systems for lab animals have been developed, with technical parameters according to the requirements of lab animals. For example, microCT and preclinical ultrasonography are now able to reach spatial resolutions as 5 um [9]. These new capabilities make these systems accessible for almost any research field involving lab animals such as oncology, infectious diseases, metabolism, orthopedics, and so.

The development of new ultrasound technologies in the last decade made accessible the scanning of lab animals almost independent of the patient size. Even mouse fetuses can be scanned with an acceptable quality in both spatial and temporal resolution. Moreover, new ultrasonographic modes have been developed, in addition to the classic A, B, M, and Doppler. In this way, now we have elastography, contrast, 4D, digital RF, or oxy-hemo modes [10]. All of them are based on the same principles as the former modes, the analysis of ultrasound waves during their travel through the tissues, but from them, we can obtain different and valuable data, such as tissue stiffness, micro-vascularization of the tissues, oxygen and hemoglobin levels, and other physiological data [10].

In this review, we will focus on the use of ultrasonography for interventionism and its potential to reduce the manipulation of animals and improve their welfare. Almost all these procedures are carried out using the B mode (brightness mode, the most typical ultrasound mode), and at some points, the Doppler mode will be useful for distinguishing the regional vascularization in order to avoid accidental punctures of the vessels. The other ultrasonography modes are, in theory, not necessary during the interventionism. The effect of ultrasound-guided surgery on the welfare of the animals could be evaluated by looking at the manipulation required for developing the same cancer model with or without this technology. We can use a pancreatic cancer model as an example. Without using an ultrasound device, we would need to anesthetize the animal, remove the air from the left lateral abdomen, and disinfect the surgical area. This procedure will be the same for the guided injection. But then, without the ultrasound, we would need to make an incision in the skin, followed by the abdominal wall. We would manually move the bowels for reaching the spleen. Later, we would expose this organ followed by the pancreatic tissue. Then, out of the body, we will inject the cells into the tissue and would put the pancreas and spleen back into the abdomen. We would close the abdominal wall and skin with surgical sutures. The whole procedure could take, for an experienced surgeon, at least 5 to 10 min and we would need to use supportive analgesia at least for the following three days. Using ultrasound guidance, after disinfecting the skin, we will scan the abdomen, find the pancreatic area, introduce transabdominally the needle into the pancreas, inject the cells, and remove the needle. Analgesia will be required for 2 days. The time for doing the injection would be 1–2 min for an experienced user. In our opinion, it is clear the effect of this technique on the welfare of the animals and projects.

Because the aim of ultrasound-guided interventionism is to introduce, or remove, material in the tissues, a minimum image quality is required for the procedures. There are no specific values for that and all depends on the target structure. For example, the required spatial resolution for a liver injection is going to be lower than the one required for a thyroid injection in the same animal; basically, because the liver is thousands of times bigger than the thyroid glands and the difficulty of injecting correctly in the organ will be low. For this reason, in this example, we do not need an extremely high spatial resolution, only sufficient to distinguish between abdominal organs. In the same way, works temporal resolution. It is not necessary to have a smooth movement of the organs in the monitor, while injecting or taking a sample, only the minimum is required to synchronize the movement of the needle and its screen visualization.

As we said, there is no optimal system or minimum requirements for doing guided injections. It will depend on the animal model (mice, rats, ferrets, rabbits…) and the structure we need to poke. For example, for injecting into the liver of a rabbit, a clinical ultrasound device with a 7.5 MHz probe will be enough, but for doing the same injection in a 20 gr mouse we will need, at least, 20 MHz, which means that almost none of the clinical devices could be used (there are some dermatology probes that could reach high frequencies such as 20 MHz) [11]. For a mouse thyroid gland or a fetus injection, the requirements reach 35–40 MHz, with values restricted to preclinical devices.

Regarding the anesthesia of the animals, it is mandatory in order to work in proper conditions and follow the minimum ethical requirements. There are several publications that show ultrasonographic examinations with awake and immobilized animals, even trained ones [12], but the interventionism required the introduction of a needle in the body, so some pain is going to be produced. Furthermore, for correct recognition of the target, we will need steady images, without movement artifacts that will complicate the synchronization of the needle movement and the ultrasound images. There are different anesthesia protocols available in the literature, with pros and cons, but for regular injections, an inhaled anesthesia would give an acceptable unconsciousness level, knowing that the pain generated in the procedure will be minimum. This level can be reached using isoflurane with an individual mask and a concentration of 2–3% of isoflurane in fresh air or oxygen. A heating platform will be required in order to compensate for the loss of temperature caused by the anesthesia. A heat light would work too in most cases. In the same way as the temperature, a lubricant must be applied to the eyes for avoiding keratitis due to dry eyes.

An analgesic protocol is required for the procedure. It is recommended to use local anesthesia at the point of injection as well as long-term analgesia during the first 48 to 72 h after the injections. This is especially required when the punctured structure is a hard organ such as the heart, liver, kidney, or spleen. In these cases, the tissular damage caused is higher than the one created when injecting a soft structure such as the pancreas. A standard protocol could be the use of bupivacaine or lidocaine as local anesthetics and buprenorphine as an acute analgesic in the first 24 h, followed by meloxicam or carprofen during the next 48 h [13]. It would be possible to inject local anesthetics in the target organ, but this will require switching syringes between the one containing the anesthetic agent and the one containing the cells. This change could be complicated if we are performing a free hand injection. Using a support for the probe and the syringes could make this change more feasible.

The needle width is a key point during the injection or the extraction of the sample. It should be big enough for the correct internal fluid displacement but the smallest to reduce the damage caused in the tissue. Therefore, the viscosity of the injected substance has an impact on the welfare of the animals and we should try to reduce this viscosity as possible. The injections can be performed manually, with regular needles and syringes, but different microinjectors can be found in the market, specifically designed for ultrasound-guided procedures [14,15]. These systems are able to set low injection volumes, down to 2 nanoliters (Figure 1).

The technical procedure to inject a liquid (drug or cells) and to remove a sample is the same but the final step will be obviously different. While doing an injection, we will push the plunger, during a biopsy we need to pull the same part of the syringe, creating an internal vacuum inside the syringe. We will need to repeat this pulling several times to be sure that we have enough content for the required analysis. This procedure is called Fine needle aspiration or FNA [16]. In hard tissues, multiple punctures can be performed without removing the needle from the organ, only changing slightly its position. With these movements, we will create more damage, but on the other hand, we will be able to sample different parts of the organ and obtain more representative samples.

## 3. Ultrasonographic Interventionism

In this section, we will describe the procedural protocols that are applied for the most frequent guided injections or samplings. From cranial to caudal, we will describe the thyroid and intracardiac injections followed by the intrabdominal organs (liver, spleen, pancreas, kidney, uterus) and end with the intratesticular injection. We will try to describe in detail the manipulation of the animal and the probe and syringe, with comments about the injected volumes and potential complications of the procedure.

It is important to say a few words about one of the main limitations of ultrasound-guided injections, and general ultrasound: it is the operator dependency of the technique. Unlike other imaging techniques, in which the analysis of the images can be performed after the acquisition, ultrasonography is a real-time technique. During the exam, we shall decide about different examination aspects, such as specific movements of the probe, increases in the probe pressure, changes in the wave frequency, brightness and contrast of the images, and so. Because of these, the results obtained during the exam are going to be affected by the expertise and experience of the operator. This is even exacerbated in ultrasound-guided surgeries, where we need to synchronize the movements of the probe and the syringe. Even if we fix the probe and needle in the respective supports, the visualization of the needle in the monitor will depend on slight and smooth movements and the probe/needle and this could be hard to achieve without previous experience and practice. For this reason, it is required to make a training before starting the real experiments and procedures with the ultrasound. The practices can be run at the beginning using carcasses for refinement and reduction purposes. As with any other surgical ability, the learning curve will depend on our skills and practice, so it is strongly recommended to repeat the procedures several times before starting the real projects.

Even with the operator-dependency limitation, the reproducibility of the models can be significantly improved using guided surgery. The reduction in the number of animal manipulations as well as the improvement in animal welfare will affect the reproducibility of the whole project, due to the lower side effects we will have from the tumor cells implantation as well as the reduced anesthesia times and improved recovery of the animal. These changes would be reflected in more homogeneous tumor developments through the group of animals. This improvement in reproducibility will have an impact on the quality of the research and animal welfare due to the reduction in outlier animals which will mean a reduction in the total of used animals [17].

### 3.1. The Thyroid Gland

The thyroid glands are two small ellipsoid and hypoechoic structures located in the ventral aspect of the neck, at both sides of the trachea, and surrounded by the salivary glands and sternohyoideus and sternothyroideus muscles. Other regional structures include the common carotid arteries and the internal jugular veins. These vascular structures make thyroid puncture significantly risky for non-expert practitioners. Even so, this risk can be reduced using the ultrasonographic Doppler mode for distinguishing the vessels from the other structures.

For intrathyroidal injections, a ventral approach is required, positioning the mouse in ventral recumbence and removing the hair of the neck and cranial part of the thorax. The front limbs are fixed with tape in a caudal position, close to the ribs. Intubation of the animal is not required and anesthesia can be supported with a facial mask. After the application of ultrasonographic gel, the scan starts locating the trachea in transversal view at the hyoid bone level. It is recognized due to the acoustic shadow produced by the intratracheal air. Moving the probe caudal, the salivary glands will appear as two superficial, bilateral, hypoechoic, and big structures. At this level, we will need to slightly increase the pressure of the probe against the neck for improving the visualization of deeper structures. Going caudal, a muscular band will appear in the middle line, ventral to the trachea, followed by two bilateral structures, the sternohyoideus and sternothyroideus muscles. The carotid arteries and jugular veins will be visible at this point. The first ones are smaller but have a pulse. We can check the blood flow direction using the Color Doppler mode of the ultrasound system. In a standardized position of the probe (left side of the probe placed over the right side of the animal), the arterial flow should be colored in red and the jugular veins should appear in blue (Figure 2).

The thyroid glands will be located at this level, dorsal and slightly medial to the neck vessels. In a standardized exam, they will appear under the vessels. They are composed of soft tissue, so the echogenicity will be lower than the salivary glands but higher than the vessels. Their shape is irregularly ellipsoidal. The best approach for the puncture is lateral, placing the needle under the ultrasound probe. If the needle is placed correctly in the injection support, we will see it coming from the lateral of the screen (Figure 2). The hardest part of the injection is piercing the skin and for this purpose, we can use forceps for immobilizing the skin. The maximum volume we can inject is low due to the organ size, so more than 10 to 20 microliters in each gland is not recommended [18,19,20].

The needle can be slowly removed after the end of the injection and a final revision of the gland is required for confirming the absence of hemorrhages.

The major risk for this technique is the injection in the wrong structure, such as the salivary glands or the regional muscles. In this case, we will see a fluid accumulation in any of the cited structures. Another risk is damage to the carotid or jugular vessels. In this case, we will see an acute hemorrhage in the zone, with a fast separation of the lobes of the salivary glands and a local swelling visually noticeable.

The thyroid biopsy should be performed in the same way as the injection, but the movement for doing multiple samplings is not recommended due to the small size of the organ and the close proximity of relevant structures such as the neck vessels or the trachea, that will have a fatal result if damaged.

### 3.2. The Heart

Intracardiac injection is a frequent task in cancer research for developing general metastatic models. Most of the time it is performed without ultrasonography and there are several manuals that describe the protocol for this injection [21,22,23,24]. The advantage of using ultrasonography is to confirm the success of the injection, since we will be able to see the needle coming into the left ventricle and the injected fluid going into the blood torrent.

The animal should be placed in a ventral recumbency and the thorax must be shaved. The probe should be placed in a transversal view over the middle part of the thorax where we will see the heart beating. In a standardized position of the probe, the left ventricle of the heart should be visible on the right side of the monitor and the right ventricle, small in comparison, will be visible on its left, mostly hidden by the acoustic shadow coming from the sternum. The approach for the injection will be from the left side of the animal, where the heart is in close contact with the inner thoracic wall. We will need to avoid the rib bones during the needle introduction into the thorax.

The less traumatic place for injection is the medial part of the left ventricle, right after the papillary muscles which can be observed as pyramidal structures growing inward from the myocardium. At this level, the needle will be more difficult to intrude, due to the muscular thickness, and the damage will be higher. Moving caudal from this point, we will find a bigger region with only the myocardial wall between the outer and inner sides of the heart. Going too caudal will lead us to the apex, where the injection will be more difficult due to its small size and thicker muscular wall.

Once we find the correct spot for injecting, the needle can be introduced from the left side avoiding the rib bones. The movement of the needle needs to be slow, and the thoracic wall can create some resistance that can be exceeded making some external pressure from the contralateral side, pushing with one finger from the right side of the thorax.

The needle can be seen inside the left ventricle as a hyperechoic linear structure with reverberation artifacts in an anechoic background. During the injection, we will be able to see some small hyperechoic dots coming from the needle. These are microbubbles created during the needle filling. These dots will confirm the correct injection in the anechoic ventricular cavity. Once the injection is completed, the needle can be removed. An example of an intracardiac injection can be seen in Figure 3.

The duration of this process depends on the expertise of the user. An experienced ultrasonographer can do the injection in less than a minute. The major risk of this procedure is the incorrect injection in the right ventricle, the lung, or the mediastinum. We will not see the hyperechoic bubbles arising inside the left ventricle. Other less frequent errors can be damaging the aorta the cava vein or any of the cardiac atriums. In these cases, we will see an acute intrathoracic hemorrhage.

For intracardiac blood sampling, the approach and procedure are the same as the injection. Sampling the myocardial tissue will be really challenging due to the thin wall and the continuous movement of the heart. We will need to puncture the cardiac wall without getting into the ventricle and make the pulling of the plunger in synchrony with the movement of the ventricle wall.

### 3.3. The Liver

The liver is a homogenous organ situated in close contact with the diaphragm, occupying the cranial part of the abdomen. Echographically, it can be described as homogeneous and moderately hyperechoic compared to the spleen [25,26,27].

The intrahepatic injection is an easy procedure due to the size of the liver, which allows us to inject in both left and right sides of the organ. In our opinion, the right approach is easier due to the presence of the stomach on the left, which reduces the space for maneuvering. The animal is placed in ventral recumbency and hair is shaved in the cranial part of the abdomen. After localizing the desired region of the liver, the needle is moved under and parallel to the probe from the outside and into the abdomen, avoiding the rib bones. We will see a hyperechoic line going into the hypoechoic and homogeneous hepatic tissue. The injection can be confirmed with the appearance of an anechoic structure (the injected fluid) inside the liver tissue. After injecting, the needle should be kept in place for some seconds. Later, it can be removed and the organ should be examined for the presence of hemorrhages. An example of a liver injection is shown in Figure 4.

The recommended maximum volume of injection depends on the size of the organ, but in the literature, we can find volumes around 40–50 microliters [28,29,30]. An excess in the injection volume could lead to a rupture of the hepatic tissue due to a pressure increase in the tissue and this will lead to the appearance of an acute local hemorrhage, or even an hemoabdomen in case the rupture affects the Glisson’s capsule. 

There are no major risks with this procedure apart from the injection of an excessive volume. Other infrequent problems could be the puncture of the gallbladder or an intrahepatic vessel. In both cases, the probability is significantly low and it could be really hard to confirm any of these situations with the ultrasound. There will be general and unspecific symptoms such as intraabdominal bleeding or peritonitis due to the release of bile into the abdomen but both findings will be delayed in time.

Making a liver biopsy will be as easier as doing an injection, following the same approach and procedure. We will be able to make multiple aspirations of the organ due to its size.

### 3.4. The Spleen

The spleen is a hypoechoic hematic organ, typically located on the left side of the abdomen, caudal to the stomach and lateral to the left kidney, but it can slide around the cranial part of the abdomen, especially during splenomegaly [25,26]. Due to this reason, the approach for its injection will depend on where it is located. In its usual place, a lateral approach is the best and easiest way of injecting. The animal should be placed in lateral recumbency, with the left side up. After shaving the hair, the scanning probe is placed over the last ribs and slowly displaced caudally. The spleen will appear on top of the screen, just under the skin. We will slightly balance the probe ventrally without losing sight of the structure and will insert the needle from the back of the animal. In a standardized view, the needle will arise from the right side of the screen and go medially. If the pressure from the ultrasound probe is enough, the spleen will be immobilized between it and the needle, and the injection will be performed easily. Similar to the liver, the maximum injected volume depends on the organ size but in previously published work we can find a range from 20 to 50 microliters [31]. After a few seconds, the needle can be removed and a last exam for the absence of bleeding should be performed. A representative image of the injection is shown in Figure 5.

When the spleen is located in a different area of the abdomen, we will do a lateral approach for the injection, placing the scanning probe in a ventral view and injecting from the left or right sides depending on the location of the structure. 

For a splenic biopsy, the approach is similar and we will be able to make several punctures from the organ due to its size in most of the animal models.

There are no big risks during a splenic injection due to its superficial location and the size of the structure. Only in immunodeficient animals, where the organ is extremely small, the risk of injecting incorrectly in another structure or even freely into the abdominal cavity should be considered. In these cases, increasing the spatial resolution for more detailed visualizations is required. The organ could be found under the last ribs and this will make the injection more complicated.

### 3.5. The Pancreas

The pancreatic tissue is, in mice, a poorly defined structure divided into three branches. The left branch is located cranial to the left kidney and medial to the spleen. It can be easily found surrounding the principal splenic vessels. The middle branch is located between the caudal aspect of the stomach and the cranial aspect of the transverse colon. The right pancreatic branch is limited by the lateral aspect of the right kidney and the medial aspect of the duodenum. The ultrasonographic image of the pancreas is as an isoechoic tissue compared to the liver, with small parallel hyperechoic lines [25,26].

For pancreatic injections, frequently performed in oncology for developing pancreatic tumors, the best area for injecting is the left branch. The middle one is smaller and the right branch is more complicated to access due to the presence of the duodenum and the right kidney. For injecting in the left branch, we need to localize the primary splenic vein that runs cranially to the cranial pole of the left kidney, from its origin in the spleen to its insertion in the cava vein. The pancreatic area can be recognized as a poorly defined isoechoic area with internal parallel hyperechoic lines around the vein. In this place, the needle can be introduced from the lateral side while we keep the scanning probe in a medial position. In a standardized image, the needle will appear from the right side of the screen. As happens with other lateral approaches, sometimes we will need to make some pressure from the contralateral side of the animal to overcome the skin and abdominal wall resistance to the puncture. Once these two structures are pierced, the injection can be performed without difficulties. A small anechoic bubble arising in the pancreatic area can be observed if the injection is performed correctly. We should wait a few seconds before removing the needle as we do in other procedures. An example of a pancreatic injection is displayed in Figure 6.

The major risk of this injection will not affect the health of the animal but the success of the model. Several times we can observe, after the injection, that the fluid from the syringe is not creating an anechoic bubble but moving freely to the ventral wall of the abdomen (the top of the monitor because we positioned the animal in ventral recumbency). This happens when the injection is not correctly performed in the pancreatic tissue but in the peritoneum. In this situation, we should stop injecting and relocate the tip of the needle to another area. Otherwise, we will have an abdominal disseminated model of pancreatic tumors.

There is no limitation on the volume injected in the pancreas due to the minimum stiffness of the tissue, but in most of the publications, the injected volume range goes from 20 to 50 microliters [32,33,34].

The guided biopsy of the pancreas is extremely difficult because, as we said before, the pancreas in mice is a membranous structure and there is not a defined tissue for aspirate. In the case of a pancreatic tumor biopsy, for example, the approach will be similar, but we should find the mass prior to the introduction of the needle. In the cells injection was correctly performed, the pancreatic tumor should be located in the same region.

### 3.6. The Kidneys

The kidneys are two ellipsoid hyperechoic structures that can be found on both sides of the abdomen. The left kidney is more caudal than the right and is anatomically related to the spleen and stomach, while the right one is close to the liver and duodenum. Their echogenicity is higher than the spleen and similar to the liver, but these ratios can change between animal strains or even individuals [25,26].

The intrarenal injection can be performed in both organs and a lateral approach is recommended, with the animal in ventral recumbency and the probe placed in the middle line of the abdomen. This layout will give us access to the lateral aspect of the kidney, safe from the ilium where the renal artery and vein are located. Depending on the injection depth, we will make a cortical (more superficial) or a medullar (inner part) injection. In both cases, the needle needs to be introduced from the lateral side and will appear in the monitor from the right side (if we inject the left kidney) or the left side (if we inject the right one) as far as we follow the scanning standardization. 

A different approach for the injection can be performed with the animal in lateral recumbency, placing the probe in the ventral aspect of the abdomen and injecting from the dorsum. This will create a compression between the needle and the probe, immobilizing the organ in the middle.

The injected fluid will be observed as a hypoechoic accumulation inside the echoic renal tissue. The maximum volume that we can inject is limited to 20–50 microliters if we attend to the published works [35,36]. The renal tissue is rigid and fragile and does not accept significant increments in the tissular pressure.

In the same way as the liver and spleen, the kidney biopsy can be performed following the same approach as the injection but pulling the needle plunger instead of pushing it down. Multiple samples can be collected in both the cortical and medullar areas of the organ. An example of a renal injection is shown in Figure 7.

The major risk during a renal injection is the incorrect settlement of the fluid. If the injection is performed too deeply into the renal tissue, the tip of the needle could reach the pelvic zone. In this case, the injected fluid will be released directly into the pelvic area and moved to the ureters. Another risk of incorrect injections could be injecting into the renal vessels, especially the renal vein, bigger than the artery. But this possibility is low if we make a safe approach from the lateral or dorsum.

### 3.7. The Uterus and Fetus

The uterus is a structure located in the caudal region of the female abdomen. It’s divided into two parts: the neck and the horns, all with a similar ultrasonographic image: tubular shape with middle echogenicity [25,26]. The neck is anatomically related to the urinary bladder and the rectum, while the horns run from their division in the cranial aspect of the neck until they reach the ovaries, caudal and lateral to the kidneys. Recognizing the uterine horns can be challenging for non-experienced users, especially in non-gravid animals when the image can be mixed up with the bowels. During the pregnancy, the uterine horns increase exponentially their size to host the developing fetus and they are easily recognized.

The uterine injection using ultrasonography is one of the most difficult techniques, especially when the organ is in repose (not gravid). The structure is long, thin, and mobile, and the uterine wall is hard. All these characteristics make the intrauterine injection a challenging process. The less mobile part of the uterus is the neck which stays anatomically fixed in place. In this part of the organ, the injection could be feasible. On the other hand, for injecting the uterine horns they should be externally exposed with a surgical opening of the abdomen. For intrauterine injections, the previously published works never went over 30 microliters [37,38,39,40]. Like in the other injections, we should keep the needle in place for some seconds before removing the needle. Figure 8 shows a representative image of a uterine injection.

The injection in the gravid uterus is easier due to the increased size of the organ but can be challenging depending on our target. The myometrium will be more complicated to reach due to the reduction in its thickness during pregnancy, as well as the injection into the uterine cavity will be challenging too because of the presence of multiple vesicles corresponding to the amniotic sacs.

If we want to inject into the amniotic sacs, the feasibility of the procedure will depend on the stage of pregnancy and fetal development. During the earliest days, recognizing the sacs will be challenging due to their small size. With the progression of fetal development, the amniotic sacs become bigger and the injection will be easier. At the end of the pregnancy, the amniotic fluid is significantly reduced and almost the whole volume of the sac corresponds to the fetus [41,42,43].

Injecting a fetus could be challenging depending on the organ or structure we want to pierce. The fetuses are mobile and they will be hard to immobilize. For this reason, most of the published works make a surgical opening of the maternal abdomen, exposing the uterine horns for ultrasonography and fetal-guided injection. Once the uterine horns are externalized, the injection procedure would be similar to the one performed in a mature animal, but the recognition of the organs could be more challenging because in some cases their echogenicity is different, such as what happens with the lungs or the kidneys. Even so, some organs, such as the testicles, are located in a different place compared to a born specimen [43].

The biopsy of the non-gravid uterus will be as challenging as the injection, due to the same reasons: its size and the almost free movement of the horns. It could be easier to make in the uterine neck. On the other hand, amniocentesis can be performed in the gravid uterus if we are able to immobilize the uterine horn, but we should consider the potential damage to the fetus due to the reduction in the amnios volume which could be a significant secondary effect of the procedure.

The main risk of failure during a uterine puncture is the incorrect injection in the abdominal cavity because of an incorrect approach to the organ. Other problems will be more related to the gravid situation, such as uterine wall ruptures if we try to inject an excessive volume into the amniotic sac or damage the fetus during the injection.

### 3.8. The Testicles

The testicles are two mobile hyperechoic structures that can be found randomly inside the scrotum, in the abdominal cavity, or in the inguinal channels [25,26]. For the intratesticular injection, we should move the organs into the scrotum, where the injection can be performed easily. A constant pressure in the caudal region of the abdomen is required to keep the testicles inside the scrotum.

Once the testicle is fixed into the scrotum, the lateral approach is the best for the injection. Placing the scanning probe in a ventral position over the scrotum, the structure can be defined echographically as hyperechoic, homogenous and circular in shape. The lateral approach is the best for injecting into the testicles, placing the probe in the middle line of the body over the scrotum. For reducing the potential movement of the organ, we can use forceps for fixing the skin while piercing with the needle. Introducing the needle in the testicular stroma is easy due to its softness and it can be seen as a hyperechoic line with several comet tails artifacts in a hyperechoic background. The injected fluid will arise as an anechoic accumulation in the testicular stroma. After the injection, we should keep the needle in place for some seconds before removing it. An example of a testicular injection is represented in Figure 9.

The maximum injected volume ranges from 20 to 50 microliters if we attend to the previous publications [44,45]. The testicle is a soft and flexible tissue with a significant capability to increase its volume so the possibility of a tissular rupture can be considered low but not inexistent.

The testicle biopsy will be as easier as the injection following the same protocol. We will need to take care of the biopsy point, because a puncture too close to the epididymis will give us a sperm sample instead of real tissue.

The are no major risks during a testicle injection. Only damage to the local vessels could create a negative effect on the viability of the organ. For avoiding this, the color Doppler mode can be used for recognizing the testicular vascular network prior to the injection.

### 3.9. Other Organs

There are other structures and organs that can be a target for orthotopic cancer models, such as the different sections of the gastrointestinal system or the urinary bowel. In all of them, we will face an additional complication to the procedure and it is that all of these structures are hollow. This means that the injection should be performed in the wall of an empty structure and in most situations, we will not have the technical precision required to make the injection or aspiration. In addition, as we described in the uterine horns, most of these structures are mobile and they will slide with the pressure of the needle tip. The only fixed structures in this group could be the stomach and the urinary bladder. For the first, injecting in the gastric wall could be feasible but the visualization of the injection will be difficult due to all the gas present in this organ, especially in rodents, that will create an acoustic shadow making it almost impossible to visualize the gastric wall. Regarding the urinary bladder, the main difficulty of its injection will be the thin wall of the bladder, especially if there is urine inside. This thin wall makes the exact injection extremely difficult. On the other hand, cystocentesis (sterile urine collection) is an easy procedure that can be performed with the help of an ultrasound and only requires the presence of urine in the bladder. The approach is ventral and we will find the bladder as a complete anechoic structure, round in shape, in the caudal part of the abdomen. Once we find it, the procedure requires introducing the needle from a lateral aspect and almost perpendicular to the probe, firmly and deep into the abdomen. If we succeed in getting into the bladder, a hyperechoic spot will be visible in the anechoic structure of the bladder. An example of a cystocentesis can be seen in Figure 10.

## 4. Conclusions

Ultrasonography is a potent and trustable technique that can help investigators in developing different cancer models. Ultrasound-guided injection is a significant tool for ethical refinement which should be considered during the design of animal experiments for an improvement in animal welfare and the quality of the research.

For the implementation of the guided injection or sampling, instruments with specific requirements are necessary, depending on the animal model and target structure, as well as a deep knowledge of the procedures by the operator is required.

## Figures and Tables

**Figure 1 animals-12-03445-f001:**
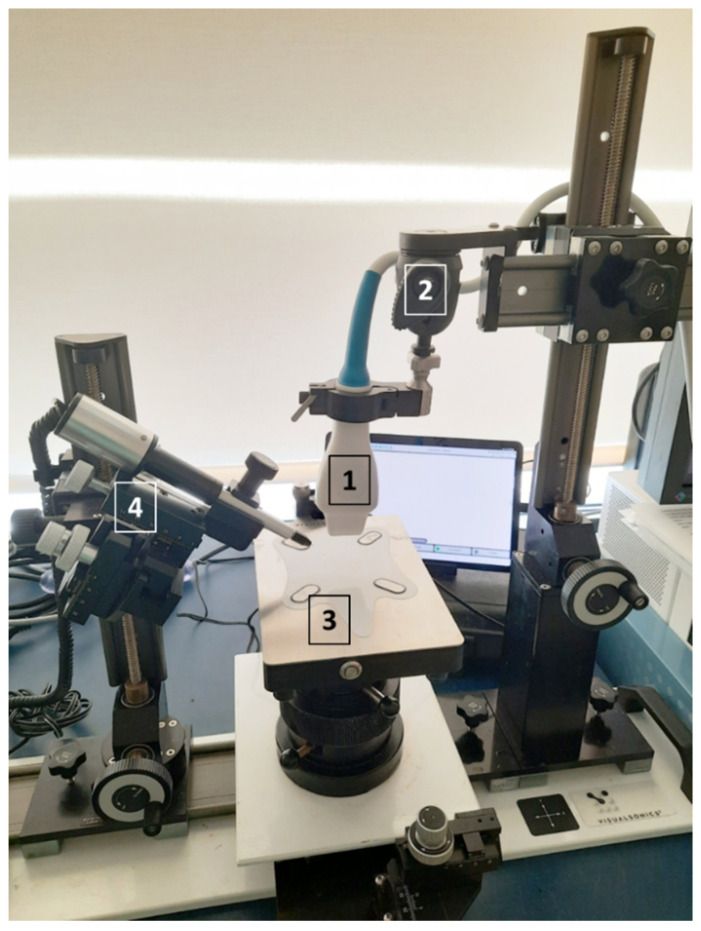
Ultrasound set with the probe (1) placed in the support (2). The heating platform for placing the scanned animal (3) and the microinjector system (4) are displayed too.

**Figure 2 animals-12-03445-f002:**
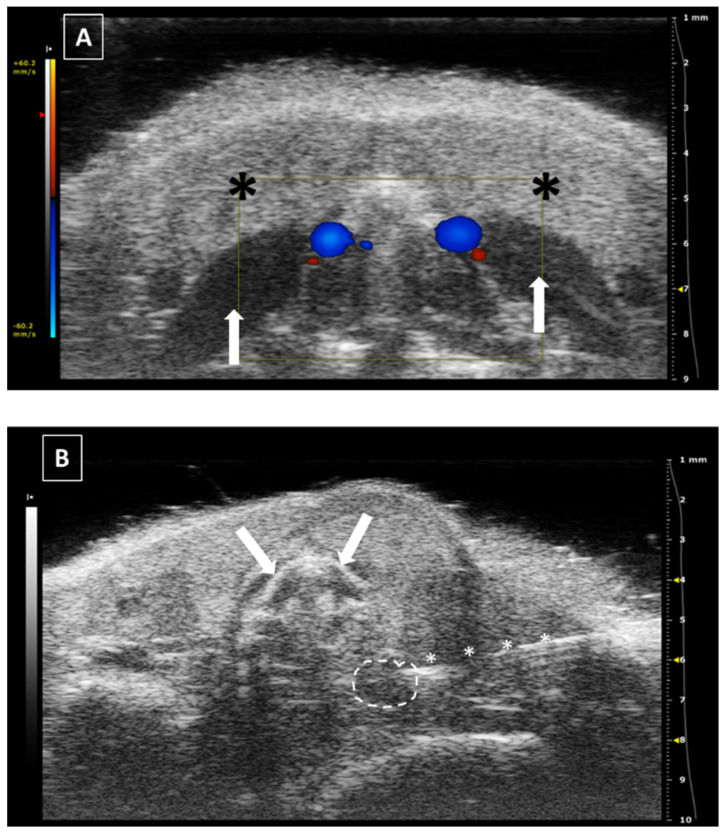
Thyroid gland injection. (**A**) Doppler mode of the medium level of the neck, where the jugular veins are colored in blue while the carotid arteries in red. Salivary glands marked with a black asterisks and neck muscles with a white arrow. (**B**) B mode during the thyroid injection. The needle is marked with white asterisks, and the thyroid gland is surrounded by a dashed line. Trachea cartilage is signaled with white arrows. Images acquired with 40 MHz frequency in B mode and 32 MHz in Doppler mode.

**Figure 3 animals-12-03445-f003:**
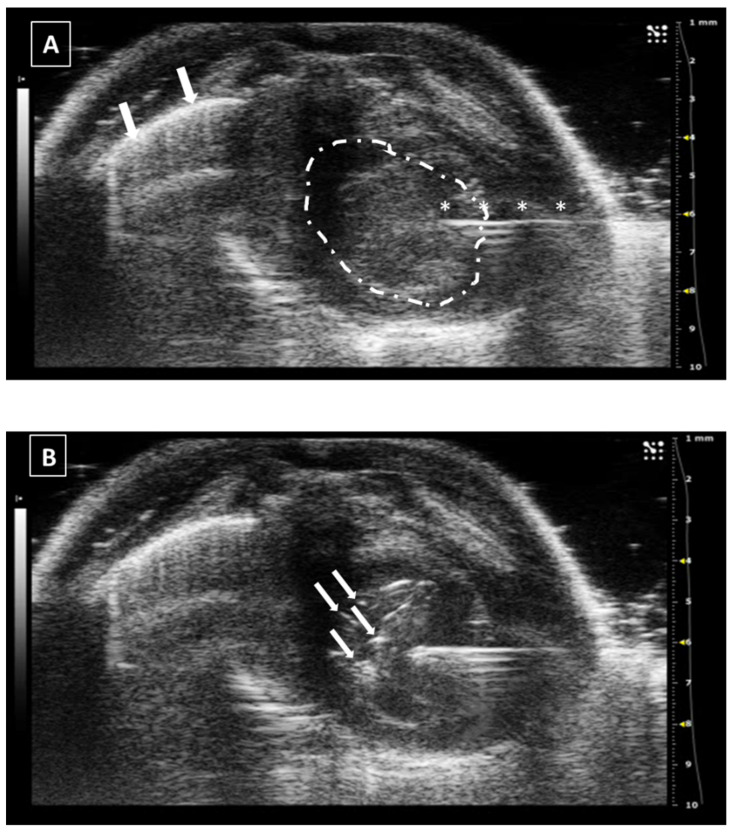
Intracardiac injection. (**A**) Preinjection image. Needle is marked with white asterisks and left ventricle area is surrounded by a dashed line. Lung artifact is labeled with white arrows. (**B**) Injection moment. Multiple white dots (marked with white arrows) inside the left ventricle correspond to microbubbles injected with the suspension. Images obtained with 40 MHz frequency.

**Figure 4 animals-12-03445-f004:**
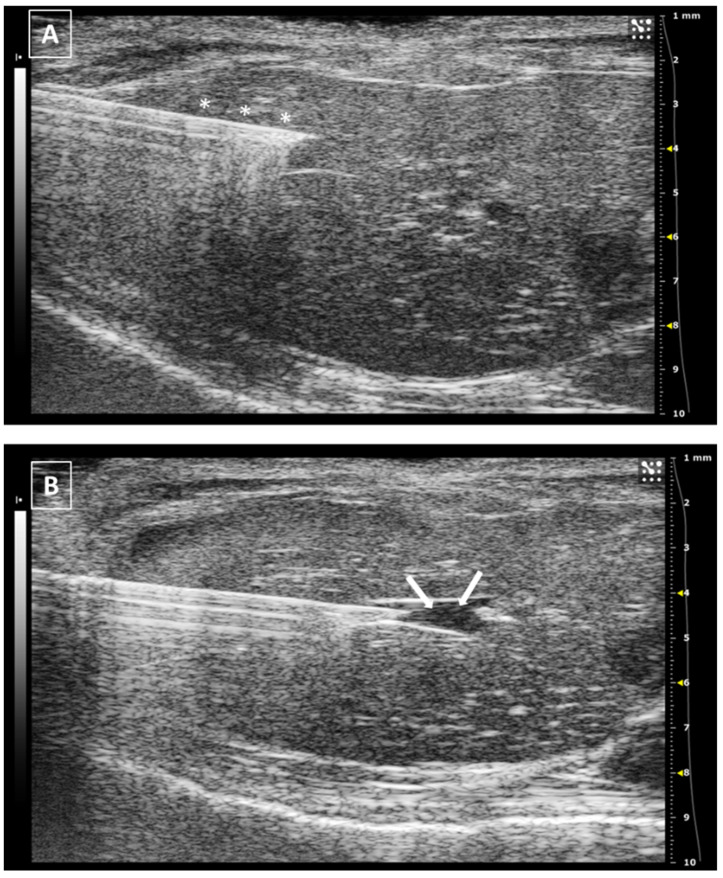
Intrahepatic injection. (**A**) Right side approach. Needle is marked with white asterisks. (**B**) Left side approach. Injected fluid is marked with white arrows. The fluid appears as an anechoic collection inside the homogeneous hypoechoic liver tissue. Images obtained at 40 MHz frequency.

**Figure 5 animals-12-03445-f005:**
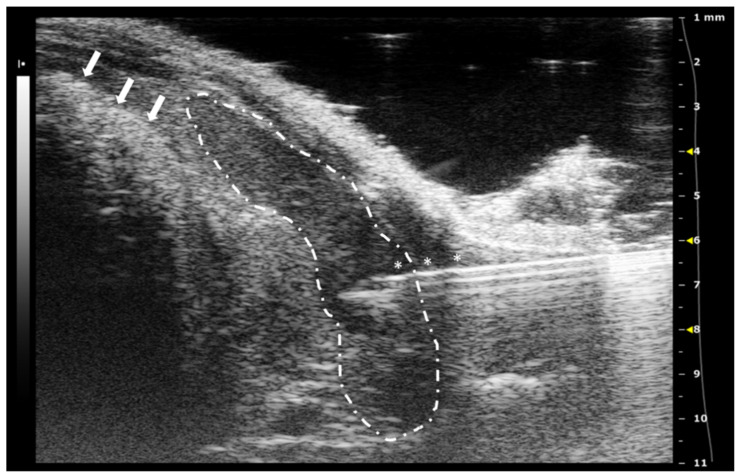
Intrasplenic injection. Needle is marked with white asterisks and the spleen is surrounded by a dashed line. Stomach can be localized due to its typical acoustic shadow. Images obtained at 40 MHz frequency.

**Figure 6 animals-12-03445-f006:**
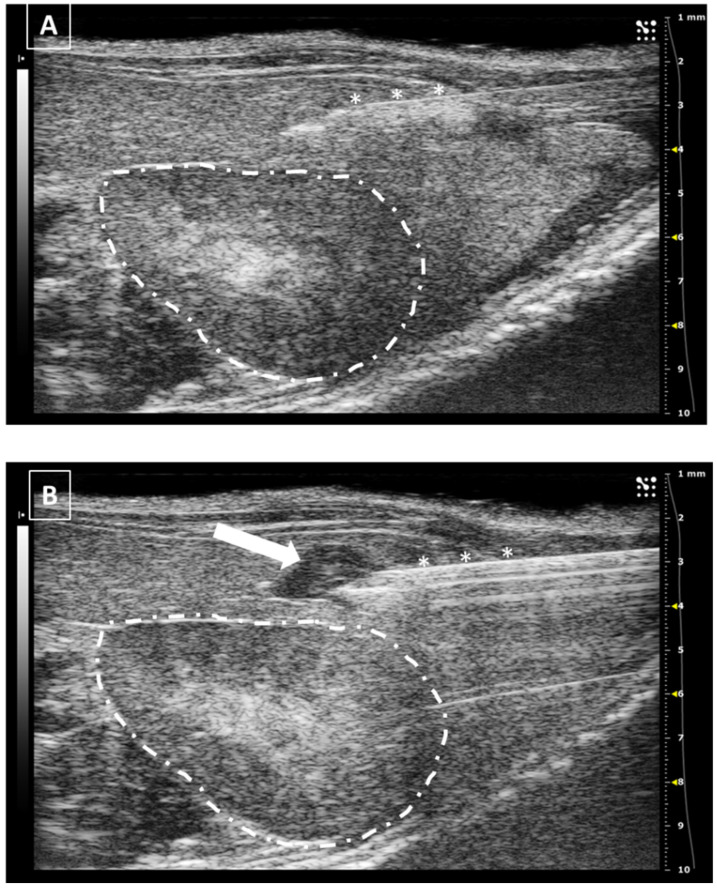
Intrapancreatic injection. (**A**) The needle is marked with white asterisks and injected in the pancreatic area. The left kidney is surrounded with a dashed line. (**B**) Same area after injection. The fluid collection is signaled with a white arrow. Images obtained at 40 MHz frequency.

**Figure 7 animals-12-03445-f007:**
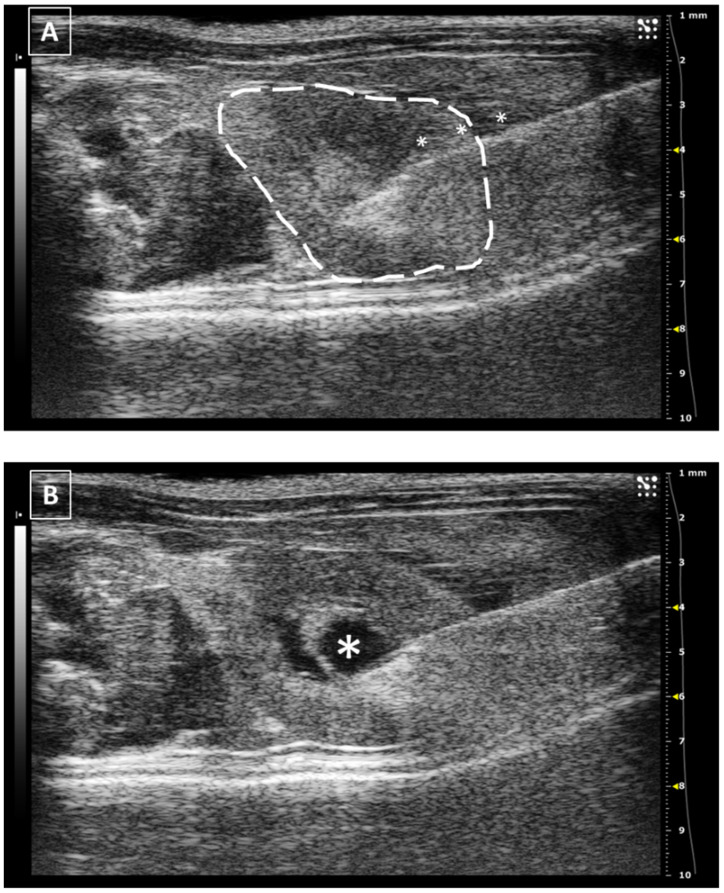
Intrarenal injection. (**A**) Injection of the needle into the kidney. Needle is marked with white asterisks and the kidney is surrounded by a dashed line. The injection is performed in the medullar zone of the organ. (**B**) Administration of the fluid, that is marked with a white asterisk. Images obtained at 40 MHz frequency.

**Figure 8 animals-12-03445-f008:**
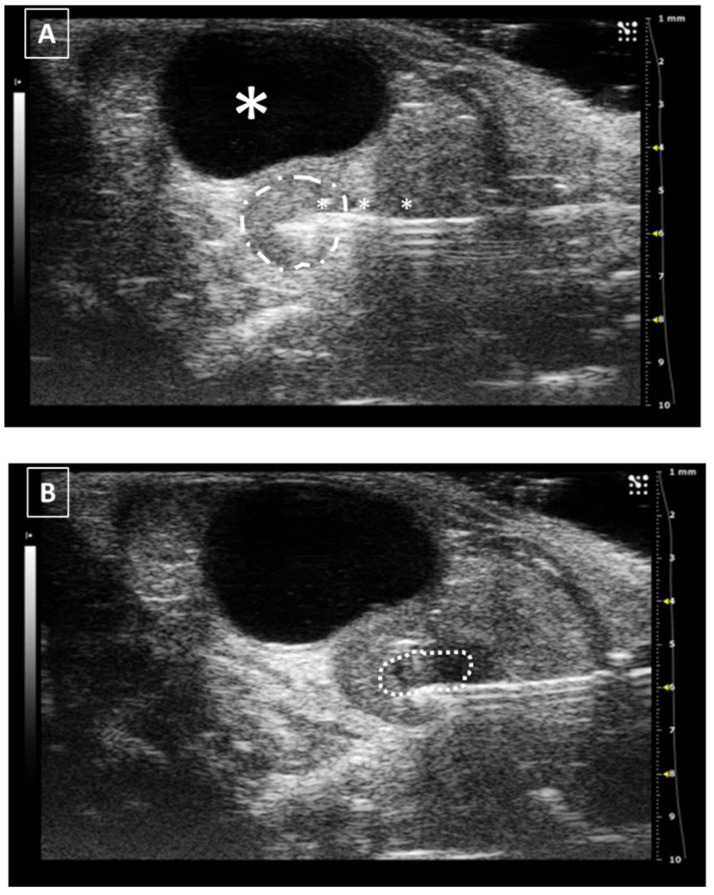
Intrauterine injection. (**A**) The uterus is punched but no fluid is administered. The needle is marked with white asterisks. The uterus is surrounded by a dashed line. The urinary bladder is marked with a big white asterisk. (**B**) Same structure after administration. The fluid collection is marked with a dotted line. Images obtained at 40 MHz frequency.

**Figure 9 animals-12-03445-f009:**
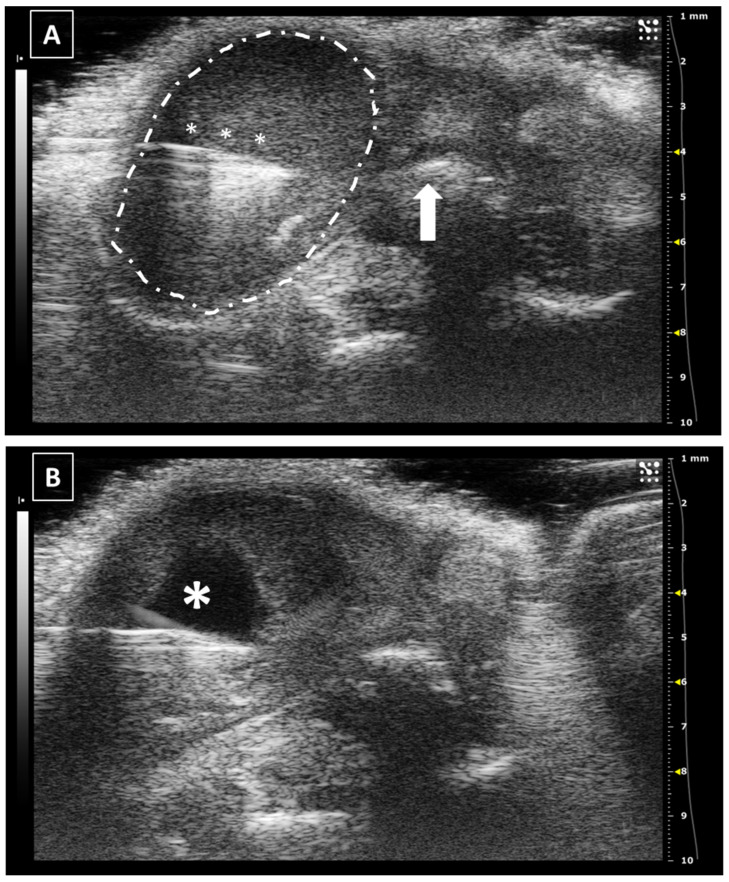
Intratesticular injection. (**A**) Injection moment. The needle is marked with white asterisks and the testicle is surrounded by a dashed line. The penis bone is marked with a white arrow. (**B**) Administration moment. The fluid collection is marked with a white asterisk. Images obtained at 40 MHz frequency.

**Figure 10 animals-12-03445-f010:**
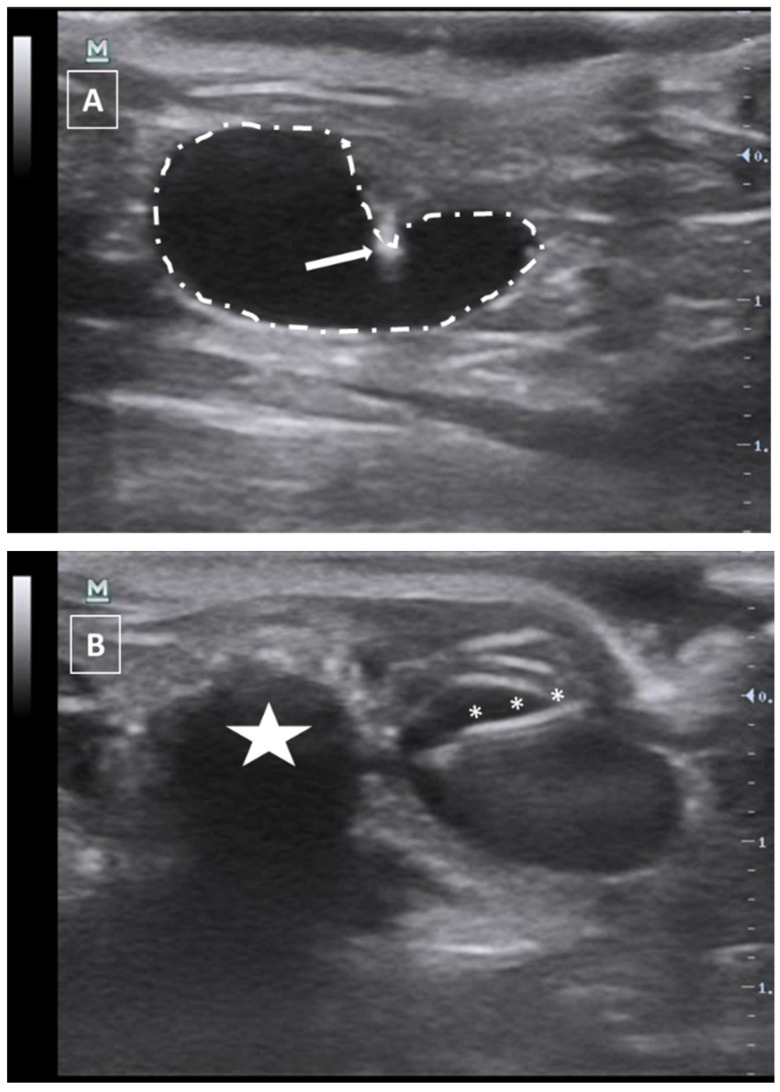
Cystocentesis. (**A**) Moment of the bladder wall rupture in transversal view. The wall is still presenting resistance and the shape of the bladder is not round due to this. The bladder is surrounded by a dotted line. The tip of the needle is marked with a white arrow. (**B**) Collection of urine. Longitudinal view. The bladder has been punctured and its wall recovered the tension, getting back the round shaper of the organ. The needle is marked with white asterisks. Cranial to the bladder, an acoustic shadow indicates the presence of feces in the rectum (white start). Images obtained at 25MHz frequency.

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
