# Peer review of "Ultrasound Guided Surgery as a Refinement Tool in Oncology Research"

_animals, 2022, doi:10.3390/ani12233445_

Round 1

Reviewer 1 Report

I consider that this article will be a very useful tool for all the researchers involved in this increasing oncologic basic research. It’s true that the refinement is the most difficult principle of the 3R’s to be done. This article has ha very importance stressing the usefulness of ultrasound to guide fine needles to do aspirations or injections avoiding unnecessary surgical interventions.

This article is systematic and well done. The figures are representatives, and the needle is very well seen in every picture, in the plan of the ultrasonographic section, the best way to do this kind of procedures.

I am convinced that this paper will help to make a very substantial improvement in basic oncological research, and I think that is ready to be published.

But I want to do some minor comments:

I think that is very important to talk about skills. To do interventional procedures guided by ultrasound is not so easy and is not so trivial. I suggest to the author to talk about that: is important to do training before to do this kind of procedures. We can guide needles in the plane or out of plane. We can do a free-hand technique, or we can fix the ultrasonographic section. Is important to talk about the learning curve and the need to practice without animals before, precisely to accomplish the 3R’s requirement.

It would be a good option to talk about some limitations because in some oncologic models is necessary to do orthotopic implants in empty viscera like stomach, bowels, or urinary bladder. In these cases is not so easy to do, now, guided by ultrasound and we need more research or more technical improvement on that to do this refinement.

I would like to ask to author if local anesthesia is important to use in some cases. 

Line 17: I suggest adding that, in human medicine, ultrasonography is currently used for guided punctures, biopsies, drainages and ablations.

Line 68: “…but IN other cases…”

Line 118: “… Doppler…”

Author Response

Thank you very much for your encouraging words and useful comments. In the new version of the document we fixed the spelling errors noted by the reviewer as well as we made a specific paragraph talking about the empty organs. Moreover, we described the protocol for the cystocentesis guided by ultrasound which we think could be interesting for the readers.

The reviewer is completely right about the skills and operator dependency of the technique, so we wrote a extent comment regarding this topic and included a new paragraph about the variability of the results, the operator effect on this variability and the global effect of both in the animal welfare.

We described more detailed the anesthesia and analgesia protocols as suggested.

Reviewer 2 Report

The review entitled „Ultrasound Guided Surgery as a Refinement Tool in Oncology Research” provides an overview on ultrasonography-based techniques for orthotopic cancer induction or biopsy collection. Due to the minimal invasiveness, these procedures potentially minimize the overall burden to the animals and improve their welfare, which is in context with the 3Rs principle by Russell and Burch.

The review give a short summary on the 3Rs, followed by a substantial overview on ultrasonography interventionism. The described techniques will potentially improve the overall wellbeing of laboratory animals during experimentation and are of great relevance for the research community.

Nevertheless, I would recommend implementing a short section summarizing the recommendations regarding animal welfare in cancer research (Workman, P., Aboagye, E., Balkwill, F. et al. Guidelines for the welfare and use of animals in cancer research. Br J Cancer 102, 1555–1577 (2010). https://doi.org/10.1038/sj.bjc.6605642). In addition, the authors should address some aspects regarding data variability and its minimization due to better animal welfare (e.g Voelkl, B., Altman, N.S., Forsman, A. et al. Reproducibility of animal research in light of biological variation. Nat Rev Neurosci 21, 384–393 (2020). https://doi.org/10.1038; Prescott, M., Lidster, K. Improving quality of science through better animal welfare: the NC3Rs strategy. Lab Anim 46, 152–156 (2017). https://doi.org/10.1038/laban.1217; …)

Line 156: please ad information on refined pain management options, e.g. multimodal analgesia. For the required needle injection, a local anesthesia with subsequent analgesia is recommended (e.g. Foley PL, Kendall LV, Turner PV. Clinical Management of Pain in Rodents. Comp Med. 2019 Dec 1;69(6):468-489. doi: 10.30802/AALAS-CM-19-000048. Epub 2019 Dec 10. PMID: 31822323; PMCID: PMC6935704.; Flecknell P. Analgesics in Small Mammals. Vet Clin North Am Exot Anim Pract. 2018 Jan;21(1):83-103. doi: 10.1016/j.cvex.2017.08.003. PMID: 29146033.; …).

Minor points

Line 32, 35 : please correct misspelling “Russel” into “Russell”

Line 68: please correct misspelling “is” to “in”

Line 116 : please considering implementing a section on less data variability due to better animal welfare (see comment above)

Author Response

Thank you very much for your comments and ideas. We have corrected the minor spelling mistakes.

We included a comment about the animal welfare guidelines and included the suggested reference. We described an example comparing a surgical approach and guided injection in a pancreatic model for showing the effect of this technology in the animal welfare. We wrote a paragraph about the variability and reproducibility and included a new reference. Regarding the anesthesia, we implemented the description of the pain management and included the suggested reference.